# Investigating the Attitudes of First-Year Students of the Faculty of Physical Education and Sports of Galati towards Online Teaching Activities during the COVID-19 Pandemic

George Danut Mocanu [1], Gabriel Murariu [2,*,†], Lucian Georgescu [2,*,†] and Ion Sandu [3,4,5]

[1] Faculty of Physical Education and Sport, "Dunarea de Jos" University of Galati, 800008 Galati, Romania; george.mocanu@ugal.ro
[2] Faculty of Sciences and Environment, "Dunarea de Jos" University of Galati, 800008 Galati, Romania
[3] Academy of Romanian Scientists (AOSR), 54 Splaiul Independentei St., Sect. 5, 050094 Bucharest, Romania; ion.sandu@uaic.ro
[4] Departamentul Interdisciplinar Științe, Institutul de Cercetări Interdisciplinare, Universitatea Alexandru Ioan Cuza Iasi, Centrul ARHEOINVEST, Bld Carol I no. 11, 700506 Iași, Romania
[5] Forumul Roman de Inventica, Str. Sf. P. Movila 3, L11, III/3, 700089 Iasi, Romania
* Correspondence: gabriel.murariu@ugal.ro (G.M.); lucian.georgescu@ugal.ro (L.G.); Tel.: +40-74-012-6940 (G.M.)
† These authors contributed equally to this paper as senior authors.

**Abstract:** The paper identifies the perceptions of first-year undergraduate students from the Faculty of Physical Education and Sports in Galati on online teaching activities, dominant and needful in the current pandemic context. The questionnaire used contains 23 items and was structured based on four distinctive factors, namely attractiveness, accessibility, motivation and efficiency; it was applied after the winter session of the academic year 2020–2021. The values of the internal consistency coefficient Cronbach's alpha indicate for the four mentioned factors a high fidelity for the measurements of the investigated features. The results of the 147 completed questionnaires allowed the definition of the independent variables sex (boys and girls) and environment of origin (rural and urban) the identification of their influence on the scores of each item (dependent variables) by using the statistical technique MANOVA (multivariate and univariate analysis), besides the values of F and the corresponding significance thresholds; the magnitude of the effect, expressed by partial eta squared ($\eta^2_p$), was also calculated. Even if the averages of item scores differ between sexes and backgrounds, the differences noted are in few cases significant: attractiveness and socialization for those in urban areas; participation in activities and effective involvement for girls; technical deficiencies, platform logging and weak computer skills for those in rural areas; and an increase in free time for girls and students in urban areas. The study undertaken allows the identification of the favorable aspects and the shortcomings of online teaching activities, these being the premises for optimizing the teaching process in the following stages.

**Keywords:** pandemic; students; e-learning; opinions; physical education

## 1. Introduction

The COVID-19 pandemic generated major transformations and challenges in the higher education system, forcing the decision-makers to quickly adopt unprecedented measures in order to carry out teaching activities in special conditions. Assuring students and parents about achieving planned competencies and insisting on asynchronous activities are goals that must be achieved in the context of promoting online teaching activities, but ways to compensate for potential shortcomings in post-pandemic training must also be identified, depending on student specialization [1]. Important changes have also taken place in physical education at the higher education level, and there are also undeniable advantages that accompany the online teaching option, namely reduced tuition costs and

increased accessibility to higher education for more people, signaling a higher number of students enrolled vs. the face-to-face variant [2]. In the case of physical education, online learning has not yet streamlined the implementation mechanisms; it facilitates access to information through various platforms/applications, but this aspect must be adapted to regional and local particularities, and maximizing the learning capacity is conditioned by the internet access [3].

In Romania, the alternative of teaching in a hybrid system is a solution that facilitates the adaptation to the problems generated by the pandemic in the higher education system, thus diminishing the percentage of the students who would react by dropping out of school. This alternative makes the transition from the classic way of teaching/face-to-face system to the one based exclusively on the scenario of online activities, avoiding negative perceptions and the shock felt when moving online [4]. Artificial intelligence can support the educational system without completely replacing the classic teaching alternative or option [5]. The strong evolution of technology facilitates the continuity of teaching in the academic world, through the existence of virtual assistants with a role in supporting the learning process [6], but the transition from classical to hybrid education is possible only by identifying new needs of the students, based on data collected from them and the provision of personalized training options [7,8].

The efficient development process or unfolding of the online teaching process is conditioned by the synergistic action of all the factors involved: students/beneficiaries, teachers and members of the administrative staff. Problems are found especially for higher education institutions that have not previously experienced the option of online teaching, and the discrepancies related to internet access and the heterogeneity of students' computer skills are causes that hinder the optimization of the teaching process [9].

An essential aspect of the online educational act is its safety and verification of the authenticity of students participating or being evaluated; the authors of [10] propose a secure system to manage the learning activity, and based on its behavior, the authenticity of the evaluated participants can be identified. An option of improving online teaching activities based on technological evolution is identified by the authors of [11]; they argue the effectiveness of mobile applications based on augmented reality/AR, finding a higher involvement of the students, positive influences upon learning and reduced negative effects generated by the lack of classical education.

The exploratory research conducted worldwide by the authors of [12] identifies the level of perception of students in 62 countries regarding the effects of the COVID-19 pandemic. The weaknesses reported are aimed at difficult tasks and limited computer skills that decrease academic performance and increase the presence of anxiety, frustration and boredom. Strengths are related to the support provided by universities and hospitals compared to other institutions. Students in Europe, those who are optimistic/confident, those who are satisfied with academic life, those in the social sciences and those with a high standard of living or scholarships perceived the measures of universities as favorable. Switching to online activities has caused more problems for students in Africa and Asia, those in the applied sciences, those with a low standard of living and those working part-time.

However, other studies indicate the fact that the effects of the pandemic are strongly felt in most European countries, with lifestyle changes and negative manifestations of behavior: increased incidence of sedentary lifestyle, increased alcohol and cannabis use, worries about the atypical existing context, the manifestation of anxious and depressive states/conditions according to [13–16] and moderate and severe states of stress and anxiety among European students reported by [17–19]. The complacency of more than half of the students in sedentary activities is reported by [20], which also identifies that this negative aspect is counterbalanced by the involvement of students in high-intensity physical demands and mental concentration and that the students especially capitalize on the information provided through social networks in order to perform various motor activities. Moreover, at the level of other continents, the teaching system had to adapt to the require-

ments imposed by the pandemic and the same mental problems, aggravated anxious states, depression and insecurity, for the investigated students [21–24].

The transition of teaching–evaluation processes to online platforms is the only measure meant to protect the academic community from COVID-19 and to allow the continuation of teaching activities as pursued priorities. Other objectives include providing food for students with international mobility and counseling students to ensure mental health [25]. The continuity in learning and the avoidance of gaps in student training have imposed various measures worldwide, with aspects differing from one country to another: online courses, recorded videos, web seminars, watching links on YouTube, video conferencing, tasks and various topics for students, TV lessons, etc. However, only the face-to-face option prevents a sedentary lifestyle, degraded physical and mental health and poor socialization problems [26].

Digital technology has penetrated strongly in the educational process, as a useful and effective teaching tool for various specializations, even in language learning, increasing student performance and diversifying learning experiences [27]. The efficiency of the teaching act in the pandemic implies fast measures aiming at increasing the computer skills of teachers, ensuring the availability of the internet connection in schools and improving the digital literacy of students, computer skills being vital in the optimal development of the teaching–learning process [28].

## 2. Materials and Methods

The purpose of the study is to evaluate the attitudes of the students of the Faculty of Physical Education and Sports regarding the features of the online teaching activities by applying a questionnaire composed of 23 items with closed answers.

*Working hypotheses*:

**Hypothesis 1 (H1).** *The applied questionnaire faithfully measures the traits to be followed, an aspect confirmed by superior and significant values of the internal consistency index Cronbach's alpha.*

**Hypothesis 2 (H2).** *The averages of the items that make up the questionnaire are different for each analyzed factor.*

**Hypothesis 3 (H3).** *There are significant differences between the averages of answers to the items between men and women.*

**Hypothesis 4 (H4).** *There are significant differences between the averages of the answers to the items between the subjects from the urban area and those from the rural area.*

### 2.1. Participants

The questionnaire was distributed online to the 177 students enrolled in the first year of the Faculty of Physical Education and Sports in Galati, but 30 of them were not included in the statistical analysis of the resulting data, either because they did not respond favorably to the invitation or because their answers were incomplete for one or more series of items. The study included 147 first-year bachelor students from the Faculty of Physical Education and Sports in Galati, representing 83.05% of the total number of first-year students and 32.81% of the total number of students of the faculty enrolled in the bachelor education program, therefore being a representative percentage for this group. Of these (84 men and 63 women, 93 from urban areas and 54 from rural areas), student specializations included physical education and sports and physiotherapy. The study was conducted with the approval of the Ethics Commission of the institution and ensuring the confidentiality of data received and processed. All surveyed students had accounts with passwords and had logged on to the Microsoft Teams platform provided by the university to participate in online teaching activities.

*2.2. The Organization of the Research*

The study was designed for cross-sectional research. The methodology used was the standard one commonly used for attitudinal investigations. The design and validation of the questionnaire took place within the Research Center for Human Performance belonging to the Faculty of Physical Education and Sports (F.E.F.S.) Galați. The questionnaire contained 23 items grouped on 4 scales or distinctive factors: the attractiveness of online activities (5 items), their accessibility (8 items), the motivation for e-learning (5 items) and the efficiency of the online teaching process (5 items). To the items with closed answers were added 3 questions with free answers, related to the advantages, shortcomings and personal proposals for optimizing the teaching activities on the online platform. The 5-point Likert scale was preferred for the items with closed answers, the quantification of these scores in the grades being mentioned separately in the statistical data presentation tables. The questionnaires were sent by e-mail to the students from the studied group, at the end of the winter session/second week of February 2021, in order to receive the data related to the perception of the teaching act and the quality of the evaluation after a semester of activity and online exam session (28 September 2020 to 7 February 2021—represents the period of the first semester).

*2.3. The Statistical Analysis of Data*

The processing and the statistical analysis of the obtained data were performed using SPSS version 24 software. Indicators regarding the measurement fidelity/the internal consistency (Cronbach's alpha in raw and standardized form) of each factor, the heterogeneity and variety of answers (Hotelling's T-squared test) and MANOVA (F) multivariate analysis with the identification of the influence of independent variables (sex and area of origin) on dependent variables (responses to the items in the questionnaire) were calculated [29–36]. The confidence interval was set at 95% ($p < 0.05$). For reasons related to the multitude of data obtained from statistical processing, the present paper does not present the influence of the age variable on the opinions expressed or the centralization and interpretation of the answers to open questions, these last 2 aspects being already published in the previous study [37].

**3. Results**

The values of the Cronbach's alpha internal consistency coefficient and the results of the equality test for the averages of analyzed items (Hotelling's T-squared test) are summarized in Table 1. It is noticed that the Cronbach's alpha values are for all four analyzed factors are higher than 0.6, considered to be the lower limit for which the fidelity of a questionnaire is accepted, so the opinions expressed are measured accurately, which confirms the first working hypothesis (H1). The results of Hotelling's T-squared test have values that confirm the second working hypothesis (H2): F1 (4.143) = 16.702, F2 (7.140) = 16.643, F3 (4.143) = 38.877 and F4 (4.143) = 94.761, all related to a significance threshold $p < 0.001$, so there is variety in the expression of responses to the items in the questionnaire and their averages are not equal.

Table 2 summarizes the values of the multivariate test, highlighting the influence of the gender variables, area variables and the combination of them upon each factor of the questionnaire. As it can be noticed, the effect of independent variables upon the cumulation of items grouped by factors does not generate statistically significant threshold values or strong effects, and there is only one case where a significant value is present (the gender variable for the motivation factor generates a $p = 0.020$ and an average effect $\eta^2_p = 0.091$, so 9.1% of the variance is explained by the influence of this variable).

**Table 1.** The values of the fidelity coefficient and the test of equality of the averages for the answers to the items of the 4 factors of the questionnaire ($N$ = 147).

| Factors | Reliability Statistics | | | Hotelling's T-Squared Test | | | | |
| --- | --- | --- | --- | --- | --- | --- | --- | --- |
| | *Cronbach's Alpha* | **Cronbach's Alpha Based on Standardized Items** | **N of Items** | **Hotelling's T-Squared** | **F** | **df1** | **df2** | **Sig.** |
| **F1** Attractiveness | 0.830 | **0.828** | 5 | 68.210 | 16.702 | 4 | 143 | 0.000 |
| **F2** Accessibility | 0.863 | **0.868** | 8 | 121.490 | 16.643 | 7 | 140 | 0.000 |
| **F3** Motivation | 0.774 | **0.774** | 5 | 158.813 | 38.887 | 4 | 143 | 0.000 |
| **F4** Efficiency | 0.811 | **0.820** | 5 | 386.995 | 94.761 | 4 | 143 | 0.000 |

F—Fisher test; df—degrees of freedom; Sig.—level of probability.

**Table 2.** Results of multivariate tests.

| Factors | Effect | $\lambda$ | F | Hypothesis df | Error df | Sig. | $\eta^2{}_p$ |
| --- | --- | --- | --- | --- | --- | --- | --- |
| **F1** Attractiveness | Gender | 0.928 | 2.147 [b] | 5.000 | 139.000 | 0.063 | 0.072 |
| | Area variables | 0.949 | 1.492 [b] | 5.000 | 139.000 | 0.196 | 0.051 |
| | Gender * area variables [a] | 0.968 | 0.906 [b] | 5.000 | 139.000 | 0.479 | 0.032 |
| **F2** Accessibility | Gender | 0.934 | 1.202 [b] | 8.000 | 136.000 | 0.302 | 0.066 |
| | Area variables | 0.931 | 1.259 [b] | 8.000 | 136.000 | 0.270 | 0.069 |
| | Gender * area variables | 0.909 | 1.695 [b] | 8.000 | 136.000 | 0.105 | 0.091 |
| **F3** Motivation | Gender | 0.909 | 2.772 [b] | 5.000 | 139.000 | 0.020 | 0.091 |
| | Area variables | 0.944 | 1.637 [b] | 5.000 | 139.000 | 0.154 | 0.056 |
| | Gender * area variables | 0.987 | 0.355 [b] | 5.000 | 139.000 | 0.878 | 0.013 |
| **F4** Efficiency | Gender | 0.928 | 2.161 [b] | 5.000 | 139.000 | 0.062 | 0.072 |
| | Area variables | 0.978 | 0.639 [b] | 5.000 | 139.000 | 0.670 | 0.022 |
| | Gender * area variables | 0.930 | 2.103 [b] | 5.000 | 139.000 | 0.069 | 0.070 |

a. Design: gender + area + gender * area. b. Exact statistic. $\lambda$—Wilk's lambda; F—Fisher test; df—degrees of freedom; Sig.—level of probability; $\eta^2{}_p$—partial eta squared.

Table 3 shows the ANOVA tests results between-subjects' effects considering the cumulative consequence of gender variables on F1 factor items/attractiveness. Table 4 shows the results of the questionnaire for the attractiveness factor using an univariate test. Table 5 shows the ANOVA tests outcomes between-subjects' effects considering the cumulative results of gender variables on F2 factor items/accessibility. Table 6 shows the outcomes of the questionnaire for the accessibility factor (univariate test). Table 7 shows the ANOVA tests consequences between-subjects' effects considering the cumulative outcome of gender variables on F3 factor items/motivation. Table 8 shows results of the questionnaire for the motivation factor (univariate test). Table 9 shows the ANOVA tests results between-subjects' effects considering the cumulative outcome of gender variables on F4 factor items/efficiency. At the level of the attractiveness factor items (Table 3) there are no values of F to be associated with significant thresholds, nor increased values of $\eta^2{}_p$ to highlight strong effects, but only values indicating weak or zero effects.

**Table 3.** Tests of between-subjects' effects—the cumulative effect of gender variables * area variables on F1 factor items/attractiveness.

| Items | Indicator | Type III Sum of Squares | F(1.143) | Sig. | $\eta^2_p$ |
|---|---|---|---|---|---|
| F1.1. The attractiveness of online activities | Gender * area variables | 0.647 | 0.605 | 0.438 | 0.004 |
| F1.2. Boredom induced by online activities | Gender * area variables | 1.952 | 2.257 | 0.135 | 0.016 |
| F1.3. Socializing in the online environment | Gender * area variables | 2.371 | 2.406 | 0.123 | 0.017 |
| F1.4. The level of stress in the online environment | Gender * area variables | 0.004 | 0.005 | 0.945 | 0.000 |
| F1.5. Participation in online teaching activities | Gender * area variables | 0.001 | 0.001 | 0.976 | 0.000 |

F—Fisher test; df—degrees of freedom; Sig.—level of probability; $\eta^2_p$—partial eta squared.

**Table 4.** The results of the questionnaire for the attractiveness factor (univariate test).

| | Scale/percent | 5 Very attractive 12.2% | 4 attractive 23.8% | 3 Medium attractive 32.7% | 2 Less attractive 29.3% | 1 Totally unattractive 2% |
|---|---|---|---|---|---|---|
| **F1.1.** The attractiveness of online activities | Indicator | Mean | Dif. | F(1.143) | Sig. | $\eta^2_p$ |
| | Boys | 3132 | 0.078 | 0.192 | 0.662 | 0.001 |
| | Girls | 3054 | | | | |
| | Urban | 3295 | **0.403** * | 5.084 | **0.026** | 0.034 |
| | Rural | 2891 | | | | |
| | Scale/percent | 5 Never 12.9% | 4 Very rare 26.5% | 3 Sometimes 43.5% | 2 Often 15.6% | 1 Always 1.4% |
| **F1.2.** Boredom induced by online activities | Indicator | Mean | Dif. | F(1.143) | Sig. | $\eta^2_p$ |
| | Boys | 3233 | −0.201 | 1.561 | 0.214 | 0.011 |
| | Girls | 3434 | | | | |
| | Urban | 3429 | 0.191 | 1.411 | 0.237 | 0.010 |
| | Rural | 3238 | | | | |
| | Scale/percent | 5 Very good 14.3% | 4 Good 34% | 3 Good Enough 34.7% | 2 Weak 13.6% | 1 Very weak 3.4% |
| **F1.3.** Socializing in the online environment | Indicator | Mean | Dif. | F(1.143) | Sig. | $\eta^2_p$ |
| | Boys | 3334 | −0.106 | 0.384 | 0.536 | 0.003 |
| | Girls | 3440 | | | | |
| | Urban | 3564 | **0.353** * | 4.235 | **0.041** | 0.029 |
| | Rural | 3210 | | | | |
| | Scale/percent | 5 Inexistent 8.8% | 4 Weak 30.6% | 3 Average 44.9% | 2 Powerful 12.9% | 1 Very strong 2.7% |
| **F1.4.** The level of stress in the online environment | Indicator | Mean | Dif. | F(1.143) | Sig. | $\eta^2_p$ |
| | Boys | 3244 | −0.056 | 0.127 | 0.722 | 0.001 |
| | Girls | 3299 | | | | |
| | Urban | 3392 | 0.240 | 2.365 | 0.126 | 0.016 |
| | Rural | 3151 | | | | |
| | Scale/percent | 5 All 19.7% | 4 Most of them 54.4% | 3 Half 17.7% | 2 Occasional 7.5% | 1 Rarely/not at all 0.7% |
| **F1.5.** Participation in online teaching activities | Indicator | Mean | Dif. | F(1.143) | Sig. | $\eta^2_p$ |
| | Boys | 3700 | **−0.316** * | 4.740 | **0.031** | 0.032 |
| | Girls | 4016 | | | | |
| | Urban | 3915 | 0.114 | 0.618 | 0.433 | 0.004 |
| | Rural | 3801 | | | | |

*—the mean difference is significant at the 0.05 level.

**Table 5.** Tests of between-subjects' effects—the cumulative effect of gender variables * area variables on F2 factor items/accessibility.

| Items | Indicator | Type III Sum of Squares | F(1.143) | Sig. | $\eta^2_p$ |
|---|---|---|---|---|---|
| F2.1. Connecting to the online platform | Gender * area variables | 0.050 | 0.073 | 0.787 | 0.001 |
| F2.2. The quality of communication in online activities | Gender * area variables | 0.968 | 1.541 | 0.217 | 0.011 |
| F2.3. The difficulty of teaching activities in the online environment | Gender * area variables | 4.606 | 3.945 | 0.049 | 0.027 |
| F2.4. Effects on financial costs | Gender * area variables | 2.122 | 2.830 | 0.095 | 0.019 |
| F2.5. Quality of home conditions in online activities | Gender * area variables | 4.116 | 5.062 | 0.026 | 0.034 |
| F2.6. Adapting to the requirements of online activities | Gender * area variables | 2.466 | 3.523 | 0.063 | 0.024 |
| F2.7. The need for support from the institution to access the platform | Gender * area variables | 3.950 | 3.332 | 0.070 | 0.023 |
| F2.8. Personal level of IT skills for working on the platform | Gender * area variables | 2.114 | 3.031 | 0.084 | 0.021 |

F—Fisher test; df—degrees of freedom; Sig.—level of probability; $\eta^2_p$—partial eta squared.

**Table 6.** The results of the questionnaire for the accessibility factor (univariate test).

| | Scale/percent | 5 Very accessible 32% | 4 Accessible 43.5% | 3 Medium accessible 21.8% | 2 Hardly accessible 2% | 1 Inaccessible 0.7% |
|---|---|---|---|---|---|---|
| **F2.1.** Connecting to the online platform | Indicator | Mean | Dif. | F(1.143) | Sig. | $\eta^2_p$ |
| | **Boys** | 3945 | −0.168 | 1.378 | 0.242 | 0.010 |
| | **Girls** | 4113 | | | | |
| | **Urban** | 4122 | 0.186 | 1.706 | 0.194 | 0.012 |
| | **Rural** | 3935 | | | | |
| | Scale/percent | 5 Very good 15.6% | 4 Good 49.7% | 3 Average 30.6% | 2 Weak 2.7% | 1 Very weak 1.4% |
| **F2.2.** The quality of communication in online activities | Indicator | Mean | Dif. | F(1.143) | Sig. | $\eta^2_p$ |
| | **Boys** | 3659 | −0.188 | 1.889 | 0.171 | 0.013 |
| | **Girls** | 3848 | | | | |
| | **Urban** | 3821 | 0.0134 | 0.959 | 0.329 | 0.007 |
| | **Rural** | 3687 | | | | |
| | Scale/percent | 5 Easy 19% | 4 Reduced difficulty 21.8% | 3 Medium difficulty 36.1% | 2 Increased difficulty 20.4% | 1 Extremely difficult 2.7% |
| **F2.3.** The difficulty of teaching activities in the online environment compared to the classic ones | Indicator | Mean | Dif. | F(1.143) | Sig. | $\eta^2_p$ |
| | **Boys** | 3307 | −0.027 | 0.022 | 0.883 | 0.000 |
| | **Girls** | 3334 | | | | |
| | **Urban** | 3426 | −0.210 | 1.268 | 0.262 | 0.009 |
| | **Rural** | 3215 | | | | |
| | Scale/percent | 5 Extremely cheap 17% | 4 Cheap 40.8% | 3 Medium costs 36.7% | 2 Expensive 3.4% | 1 Very expensive 2% |
| **F2.4.** Effects on financial costs | Indicator | Mean | Dif. | F(1.143) | Sig. | $\eta^2_p$ |
| | **Boys** | 3760 | 0.162 | 1.168 | 0.282 | 0.008 |
| | **Girls** | 3598 | | | | |
| | **Urban** | 3630 | −0.099 | 0.434 | 0.511 | 0.003 |
| | **Rural** | 3729 | | | | |

**Table 6.** *Cont.*

| | Scale/percent | 5 Fully 36.1% | 4 Largely 33.3% | 3 Decent level 26.5% | 2 To a small extent 3.4% | 1 No, they are improper 0.7% |
|---|---|---|---|---|---|---|
| **F2.5.** Quality of home conditions in online activities | Indicator | Mean | Dif. | F(1.143) | Sig. | $\eta^2_p$ |
| | **Boys** | 4011 | 0.050 | 0.104 | 0.747 | 0.001 |
| | **Girls** | 3960 | | | | |
| | **Urban** | 4075 | 0.179 | 1.312 | 0.254 | 0.009 |
| | **Rural** | 3896 | | | | |
| | Scale/percent | 5 Very fast 8.8% | 4 Fast 38.8% | 3 Medium adaptation 40.8% | 2 Difficult/slow 10.2% | 1 Very hard 1.4% |
| **F2.6.** Adapting to the requirements of online activities | Indicator | Mean | Dif. | F(1.143) | Sig. | $\eta^2_p$ |
| | **Boys** | 3398 | −0.026 | 0.033 | 0.857 | 0.000 |
| | **Girls** | 3424 | | | | |
| | **Urban** | 3526 | 0.230 | 2.518 | 0.115 | 0.017 |
| | **Rural** | 3297 | | | | |
| | Scale/percent | 5 Done that alone 25.2% | 4 To a small extent 30.6% | 3 To a medium extent 32% | 2 Largely 6.1% | 1 I could not do it without help 6.1% |
| **F2.7.** The need for support from the institution to access the platform | Indicator | Mean | Dif. | F(1.143) | Sig. | $\eta^2_p$ |
| | **Boys** | 3553 | −0.034 | 0.032 | 0.858 | 0.000 |
| | **Girls** | 3587 | | | | |
| | **Urban** | 3811 | 0.482* | 6.567 | **0.011** | 0.044 |
| | **Rural** | 3329 | | | | |
| | Scale/percent | 5 Very good 16.3% | 4 Good level 36.1% | 3 Medium level 40.1% | 2 Weak level 7.5% | 1 Very weak 0% |
| **F2.8.** Personal level of IT skills for working on the platform | Indicator | Mean | Dif. | F(1.143) | Sig. | $\eta^2_p$ |
| | **Boys** | 3621 | 0.112 | 0.599 | 0.440 | 0.004 |
| | **Girls** | 3509 | | | | |
| | **Urban** | 3730 | **0.331 *** | 5.258 | **0.023** | 0.035 |
| | **Rural** | 3399 | | | | |

*—the mean difference is significant at the 0.05 level.

**Table 7.** Tests of between-subjects' effects—the cumulative effect of gender variables * area variables on F3 factor items/motivation.

| Items | Indicator | Type III Sum of Squares | F(1.143) | Sig. | $\eta^2_p$ |
|---|---|---|---|---|---|
| F3.1. Level of motivation to participate in online activities | Gender * area variables | 0.674 | 0.884 | 0.349 | 0.006 |
| F3.2. Degree of involvement in teaching activities | Gender * area variables | 0.762 | 1.057 | 0.306 | 0.007 |
| F3.3. Interest in the topics discussed at courses and seminars | Gender * area variables | 0.261 | 0.422 | 0.517 | 0.003 |
| F3.4. The effect on personal free time compared to the classic version | Gender * area variables | 0.268 | 0.405 | 0.525 | 0.003 |
| F3.5. Receiving a sufficient number of electronic teaching materials for learning | Gender * area variables | 0.142 | 0.184 | 0.669 | 0.001 |

For the items of the motivation factor, no values of F are found to be associated with statistically significant thresholds, all values of *p* being >0.05. The values of $\eta^2_p$ indicate zero effect sizes (between 0.1% and 0.7% of the variance of each item is assigned/explained by the influence of gender * area variable interaction).

**Table 8.** The results of the questionnaire for the motivation factor (univariate test).

| F3.1. Level of motivation to participate in online activities | Scale/percent | 5 Extremely motivated 4.8% | 4 Very motivated 36.7% | 3 Medium motivated 38.8% | 2 Less motivated 17% | 1 Totally unmotivated 2.7% |
|---|---|---|---|---|---|---|
| | Indicator | Mean | Dif. | F(1.143) | Sig. | $\eta^2_p$ |
| | Boys | 3089 | −0.295 | 3.814 | 0.053 | 0.026 |
| | Girls | 3384 | | | | |
| | Urban | 3332 | 0.191 | 1.595 | 0.209 | 0.011 |
| | Rural | 3141 | | | | |
| F3.2. Degree of involvement in teaching activities | Scale/percent | 5 Very involved 6.8% | 4 Actively involved 42.2% | 3 Medium involved 34.7% | 2 Weakly involved 15.6% | 1 Not involved 0.7% |
| | Indicator | Mean | Dif. | F(1.143) | Sig. | $\eta^2_p$ |
| | Boys | 3280 | −0.197 | 1.804 | 0.181 | 0.012 |
| | Girls | 3477 | | | | |
| | Urban | 3476 | 0.195 | 1.760 | 0.187 | 0.012 |
| | Rural | 3281 | | | | |
| F3.3. Interest in the topics discussed at courses and seminars | Scale/percent | 5 Very interested 11.6% | 4 Interested 56.5% | 3 Partially interested 24.5% | 2 Little interest 6.1% | 1 Totally not interested 1.4% |
| | Indicator | Mean | Dif. | F(1.143) | Sig. | $\eta^2_p$ |
| | Boys | 3545 | −0.377 * | 7.701 | 0.006 | 0.051 |
| | Girls | 3922 | | | | |
| | Urban | 3742 | −0.017 | 0.016 | 0.900 | 0.000 |
| | Rural | 3725 | | | | |
| F3.4. The effect on personal free time compared to the classic version | Scale/percent | 5 A lot of free time 12.9% | 4 More free time 46.9% | 3 The same free time 32.7% | 2 Less free time 6.1% | 1 Very little free time 1.4% |
| | Indicator | Mean | Dif. | F(1.143) | Sig. | $\eta^2_p$ |
| | Boys | 3450 | −0.348 * | 6.124 | 0.015 | 0.041 |
| | Girls | 3798 | | | | |
| | Urban | 3771 | 0.294 * | 4.367 | 0.038 | 0.030 |
| | Rural | 3477 | | | | |
| F3.5. Receiving a sufficient number of electronic teaching materials for learning | Scale/percent | 5 For all disciplines 32.7% | 4 Most disciplines 55.1% | 3 Half the disciplines 4.8% | 2 Few disciplines 5.4% | 1 None of the disciplines 2% |
| | Indicator | Mean | Dif. | F(1.143) | Sig. | $\eta^2_p$ |
| | Boys | 4055 | −0.154 | 1.019 | 0.315 | 0.007 |
| | Girls | 4209 | | | | |
| | Urban | 4091 | −0.083 | 0.295 | 0.588 | 0.002 |
| | Rural | 4173 | | | | |

*—the mean difference is significant at the 0.05 level.

**Table 9.** Tests of between-subjects effects—the cumulative effect of gender variables * area variables on F4 factor items/efficiency.

| Items | Indicator | Type III Sum of Squares | F(1.143) | Sig. | $\eta^2_p$ |
|---|---|---|---|---|---|
| F4.1 The usefulness of online activities | Gender * area variables | 0.246 | 0.263 | 0.609 | 0.002 |
| F4.2. The quality of the online teaching act | Gender * area variables | 0.011 | 0.017 | 0.895 | 0.000 |
| F4.3. Perception of the evaluation act | Gender * area variables | 0.441 | 0.548 | 0.460 | 0.004 |
| F4.4. Level of final training if the online scenario would continue | Gender * area variables | 3.049 | 3.047 | 0.083 | 0.021 |
| F4.5. Existence of other concerns while participating in online classes | Gender * area variables | 3.788 | 3.365 | 0.069 | 0.023 |



Table 10 indicates the outcomes of the questionnaire for the efficiency factor (univariate test). All these tables indicate the description of the items, the scores for each item on the Likert scale and the grade that accompanies them, separately on the four building factors of the questionnaire, as well as the percentage of students who prefer that score. The tables also present the average values by sex and area, the differences between the averages, the values of F with the corresponding significance thresholds and effect size expressed by partial eta squared ($\eta^2_p$).

**Table 10.** The results of the questionnaire for the efficiency factor (univariate test).

| | Scale/percent | 5 Very useful 7.5% | 4 Useful 38.1% | 3 Medium useful 27.9% | 2 Less useful 24.5% | 1 Useless 2% |
|---|---|---|---|---|---|---|
| **F4.1.** The usefulness of online activities | Indicator | Mean | Dif. | F(1.143) | Sig. | $\eta^2_p$ |
| | Boys | 3074 | −0.315 | 3.568 | 0.061 | 0.024 |
| | Girls | 3390 | | | | |
| | Urban | 3360 | 0.256 | 2.357 | 0.127 | 0.016 |
| | Rural | 3104 | | | | |
| | Scale/percent | 5 Very good 24.5% | 4 Good 51% | 3 Average level 21% | 2 Weak 2.7% | 1 Very weak 0.7% |
| **F4.2.** The quality of the online teaching act | Indicator | Mean | Dif. | F(1.143) | Sig. | $\eta^2_p$ |
| | Boys | 3833 | −0.233 | 2.926 | 0.089 | 0.020 |
| | Girls | 4066 | | | | |
| | Urban | 4050 | 0.201 | 2.176 | 0.142 | 0.015 |
| | Rural | 3849 | | | | |
| | Scale/percent | 5 Certainly objective 38.1% | 4 Objective 34.7% | 3 Quite objective 23.1% | 2 Weak 3.4% | 1 Very weak 0.7% |
| **F4.3.** Perception of the evaluation act | Indicator | Mean | Dif. | F(1.143) | Sig. | $\eta^2_p$ |
| | Boys | 3949 | −0.223 | 2.063 | 0.153 | 0.014 |
| | Girls | 4172 | | | | |
| | Urban | 4131 | 0.142 | 0.833 | 0.363 | 0.006 |
| | Rural | 3989 | | | | |
| | Scale/percent | 5 Very good 4.8% | 4 Better 9.5% | 3 The same 29.3% | 2 Worse 42.9% | 1 Very bad 13.6% |
| **F4.4.** Level of final training if the online scenario would continue vs. classical teaching | Indicator | Mean | Dif. | F(1.143) | Sig. | $\eta^2_p$ |
| | Boys | 2517 | 0.105 | 0.367 | 0.546 | 0.003 |
| | Girls | 2412 | | | | |
| | Urban | 2551 | 0.174 | 1.014 | 0.316 | 0.007 |
| | Rural | 2377 | | | | |
| | Scale/percent | 5 Never 20.4% | 4 Rarely 35.4% | 3 Sometimes 27.2% | 2 Often 13.6% | 1 Always 3.4% |
| **F4.5.** Existence of other concerns while participating in online classes | Indicator | Mean | Dif. | F(1.143) | Sig. | $\eta^2_p$ |
| | Boys | 3526 | −0.017 | 0.008 | 0.927 | 0.000 |
| | Girls | 3543 | | | | |
| | Urban | 3648 | 0.228 | 1.539 | 0.217 | 0.011 |
| | Rural | 3421 | | | | |

*—the mean difference is significant at the 0.05 level.

Only 2% of those surveyed indicate the online teaching option as unattractive, and at the opposite end, 12.2% find it very attractive. The differences between the sex scores are insignificant, but those between the areas are significant (F = 5.04, *p* = 0.026, $\eta^2_p$ = 0.034), so those in urban areas consider it more attractive than those in rural areas. Only 1.4% are constantly bored; boys and students from rural areas present values that indicate

higher boredom in teaching activities, but the differences are not statistically confirmed. Socialization presents balanced values by sex and insignificant differences; instead, those from urban areas get a significantly higher average score (F = 4.235, *p* = 0.041, $\eta^2_p$ = 0.029), 14.3% declaring that it is very good and 3.4% declaring that it is very weak. The absence of stress is indicated by 8.8%, while 2.7% choose the variant of very strong stress, boys being slightly more stressed than girls and those in rural areas being more stressed than those in urban areas, but the differences are statistically insignificant. Participation in activities is constant for 19.7% of students, and only 0.7% say that they participate very rarely or not at all. Those in urban areas have a slight superiority over rural ones, but insignificant; instead, girls present a significantly higher score than boys (F = 4.740, *p* = 0.031, $\eta^2_p$ = 0.032), which demonstrates a superior concern and involvement in these activities. All these important results are presented in Figure 1.

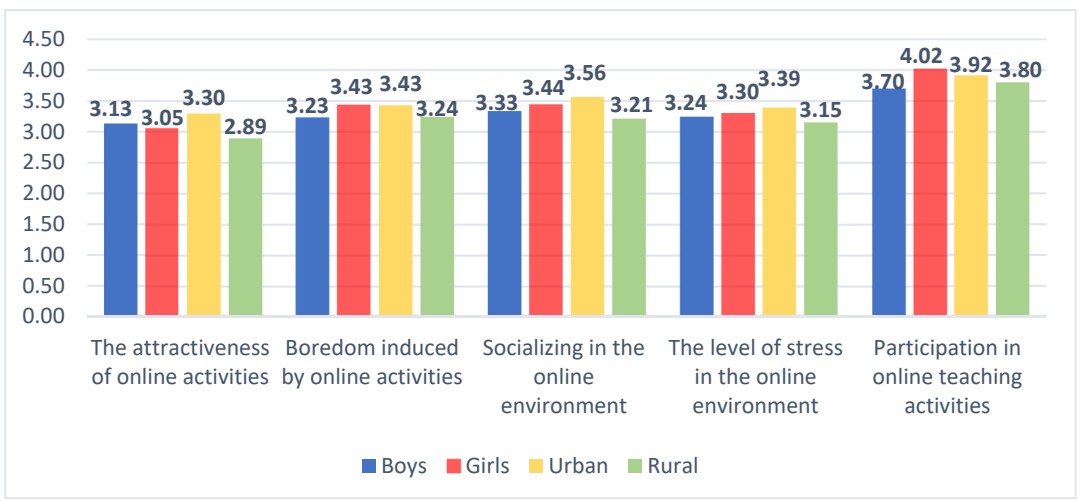

**Figure 1.** The average values of the answers for the items of the attractiveness factor.

At the level of the accessibility factor items, there are found two cases in which the interaction of the gender and environment of origin variables generates statistically significant effects, but with low values of $\eta^2_p$ (the difficulty of teaching activities in the online environment, with *p* = 0.049, and quality of home conditions in online activities, with *p* = 0.026). In the first case, for the difficulty of teaching activities in the online environment item, 2.7% of the variance is attributed to the interaction of the two independent variables, and for the quality of home conditions in the online activities item, 3.4% of the variance is explained by the same aspect/issue.

The connection on the work platform is perceived as very accessible by 32% and totally inaccessible by 0.7%; the girls and those in urban areas have higher values, but insignificant. Communication is identified as very good by 15.6% and very poor by 1.4%, with the same better and insignificant average scores for girls and the urban environment. Teaching activities are easy for 19% and extremely difficult for 2.7%, and boys and rural students have slightly more difficult environments, but also with insignificant differences. The activities are perceived as very cheap by 17% and as very expensive by 2%, but a higher average score is captured for those in rural areas compared to urban ones, even if it is still insignificant. The quality of the conditions at home is very good for 36%, and only 0.7% report improper conditions, with higher averages of boys and for those in urban areas and insignificant differences. The adaptation to requirements occurred very quickly for 8.8% of those surveyed and was very difficult for 1.4%; girls and those in urban areas presenting higher scores, but statistically insignificant. Only 25.2% of students managed to access the platform on their own, and 6.1% were dependent on the help of the faculty, gender differences being insignificant, but the score of those in urban areas is significantly higher than those in rural areas (F = 6.567, *p* = 0.011, $\eta^2_p$ = 0.044), which demonstrates the

existence of a gap in computer skills between the two areas. This aspect is confirmed by the last item of the factor, in which 16.3% believe that they have very good IT skills and no one declares that they have very weak skills, boys having a slightly but insignificantly higher average than girls, but those in urban areas have a statistically significantly higher average score than those in rural areas (F = 5.258, $p$ = 0.023, $\eta^2_p$ = 0.035). All these results are presented in Figure 2.

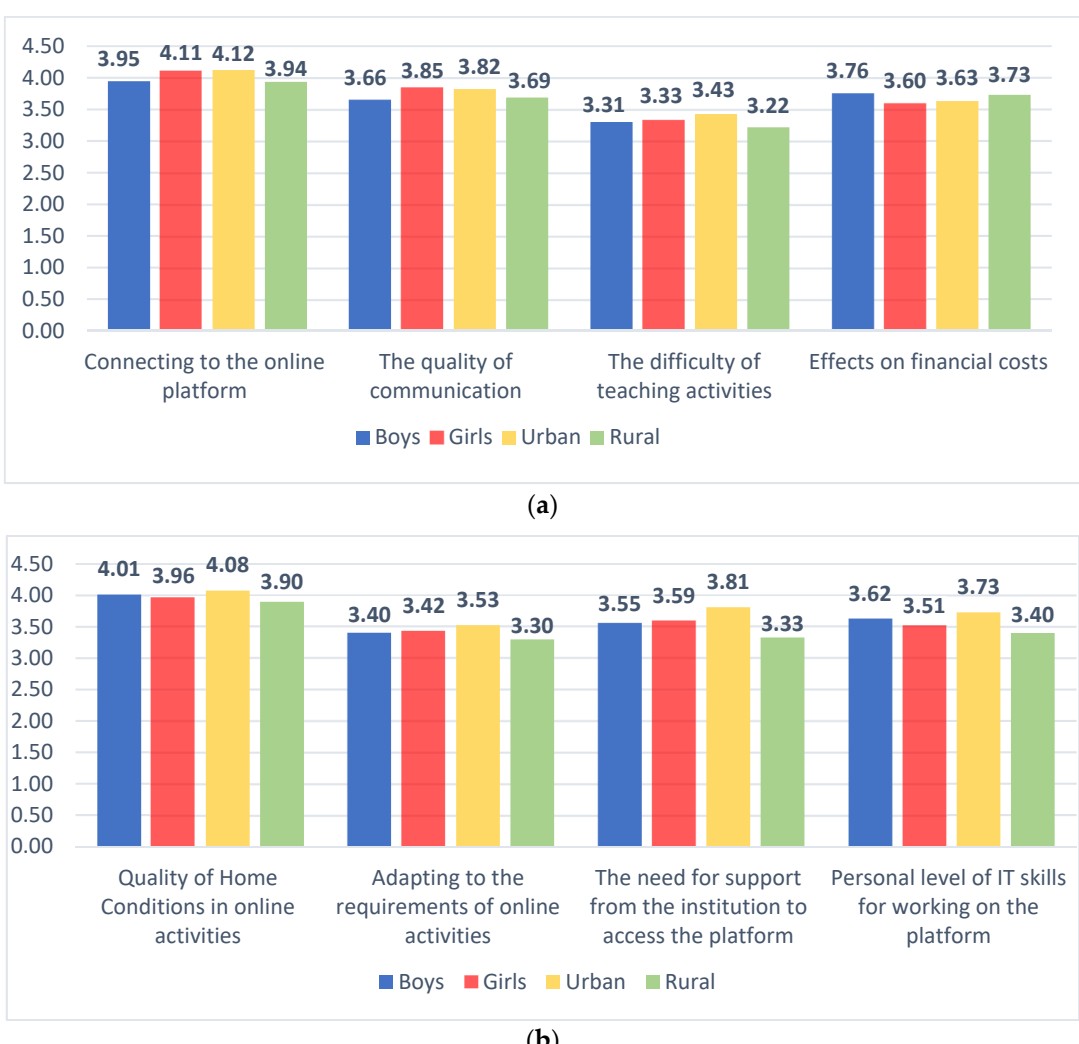

**Figure 2.** (**a**) The average values of the answers for the items of the accessibility factor. (**b**) The average values of the answers for the items of the accessibility factor.

In the case of this factor, the superiority of the average values of girls compared to boys in all five items analyzed should be noted. Only 4.8% are extremely motivated for e-learning activities, and on the other end, 2.7% are totally unmotivated. Girls present higher average scores than boys, and those in urban areas presented higher values than those in rural areas, but without significant differences. Only 6.8% say they are very actively involved in activities, and 0.7% are not involved, the differences between sexes and areas also being insignificant. Of the students surveyed, 11.6% are very interested in the topics discussed, while 1.4% are totally not interested, and the average differences between boys and girls are significant (F = 7.701, $p$ = 0.006, $\eta^2_p$ = 0.051). Online activities generate a lot of free time for 12.9% of students and a drastic limitation for 1.4%. The differences between girls and boys in this case are significant (F = 6.124, $p$ = 0.015, $\eta^2_p$ = 0.041), as are those between urban and rural students, where the urban students have more free time (F = 4.367, $p$ = 0.038, $\eta^2_p$ = 0.030). Having received enough electronic study materials for

all the disciplines studied is indicated by 32.7%, and 2% are totally dissatisfied with the number of materials received, those in rural areas being more satisfied than those in urban areas, but also with statistically insignificant differences. All these results are presented in Figure 3.

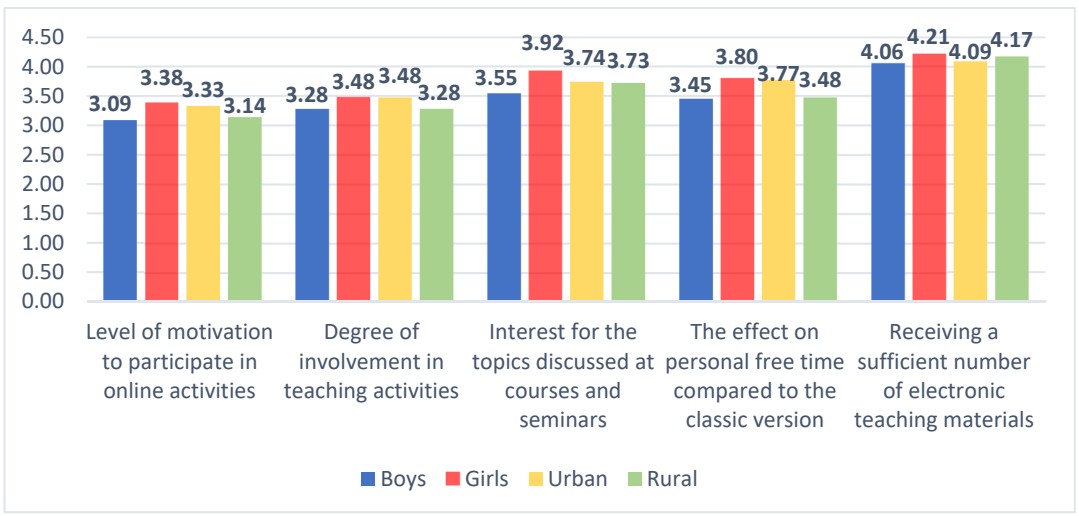

**Figure 3.** The average values of the answers for the items of the factor motivation.

At the level of the last factor, efficiency, no values of F associated with significant thresholds are found, all the values of $p$ being >0.05. For the first three items of the factor, the values of $\eta^2_p$ indicate zero effects of the interaction of the gender * area variables, and for the last two items, values of $\eta^2_p$ are obtained which indicate low effects (the interaction of gender * area variables determines 2.1% and 2.3% of the variance of these dependent variables).

In the case of the efficiency factor, it should be noted that insignificant differences between the pairs of the variables sex and residence area are obtained for all items. However, the results are relevant through the views expressed on the usefulness, the quality of teaching and evaluation, the final level of training and the assumption of behaviors that disrupt online activities. Online activities are perceived as very useful by 7.5% of students, while 2% see them as totally useless, with higher average scores for the groups of girls and urban students. The teaching act is evaluated as being very good by 24.5% of the students and as very poor by 0.7%, with the girls and the urban environment having higher average scores. The evaluation is identified as certainly objective by 38.1% and very weak/subjective by 0.7% of students, the average values being also higher for girls and students in urban areas. Continuing online training would generate a higher level of final training than the classic mode of teaching only for 4.8% of cases, and 13.6% identify the danger of exclusive online training, appreciating the final level of professional training as very low; men and those from urban are more optimistic, as evidenced by the higher average scores obtained. Regarding other concerns that overlap with online teaching activities (social networks, listening to music, watching movies, driving, physical training, work-related activities, etc.), 20.4% state that they do not have such concerns, but 3.4% and 13.6% indicate that this happens constantly and often, respectively, which raises an alarm signal regarding the ability of students to focus on the teaching tasks in this type of teaching–learning. All these results are presented in Figure 4.

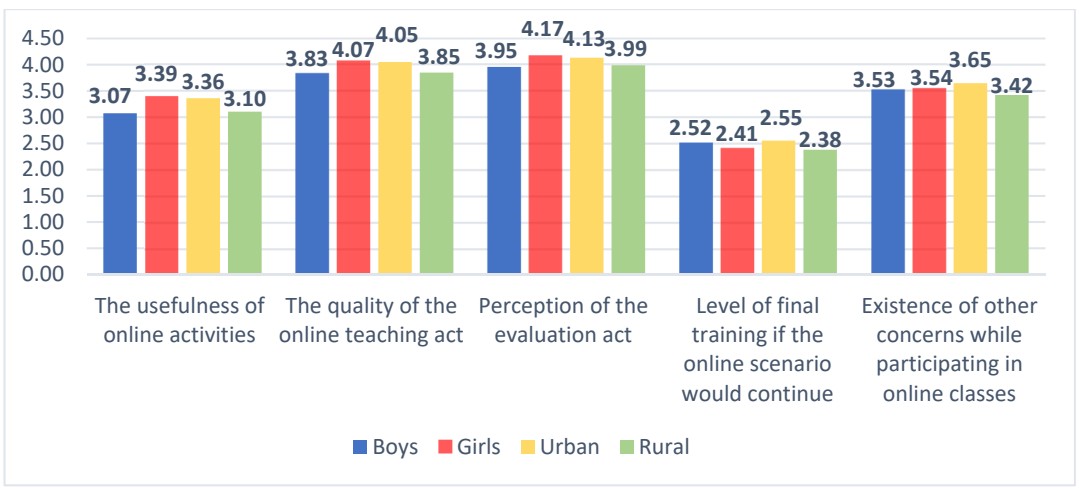

**Figure 4.** The average values of the answers for the items of the efficiency factor.

## 4. Discussion

The results of the study are broadly consistent with studies published in the literature from the beginning of the COVID-19 pandemic to the present. Actually, our study confirms the existence of gender gaps and especially gaps between students' backgrounds, allowing the identification of problems in the online teaching system in terms of internet connection, technical problems, stress and boredom associated with this teaching option, poor socialization, low concentration, the existence of other concerns, etc. On the other hand, the advantages/benefits that students identify, notably more free time, should not be neglected. The significantly lower attractiveness of online activities for students in rural areas necessitates finding solutions to improve this factor, especially since they also have scores that indicate a greater boredom than students in the urban environment, being more stressed and notably presenting slightly lower levels of involvement. The fact that girls have significantly higher scores for the factor of participation in teaching activities demonstrates a higher responsibility and a better awareness of the importance of the teaching act; they also have higher scores in the concern for the topics discussed in class. The superior socialization of those in the urban environment can also be explained by familiarization with IT equipment and better computer skills. The financial accessibility of online activities is better for boys, who perceive them as cheaper, but girls perceive a better quality of online communication. There is equality between girls and boys in the perception of the difficulty of the online teaching act vs. the classic version, but rural students feel it is more complicated/difficult and also have a problematic adaptation to the requirements of online activities, being more often forced to use the educational institution, colleagues or teachers to access the work platform and operate its functions. The cancellation of the daily commute for those in rural areas did not lead to very good scores for the amount of free time, a justification being the constant involvement in household activities, while urban students (especially girls) presented high value for this parameter. Girls and those in rural areas are more satisfied with the teaching materials provided by teachers through the work platform, and boys and those in rural areas see less usefulness of the online teaching form, as well as a lower quality of the pre-teaching process and of the evaluation act, without any significant differences. The fact that over 50% of respondents indicate a weaker or very poor training in the option of continuing online teaching activities demonstrates the limits of this compromise situation and entails the need to improve the teaching–learning–assessment process in this formula. The problem of parallel activities during the participation in online classes (in the situation where the cameras and microphones are turned off) is recognized by some students and at the same time is the main deficiency signaled by the faculty teachers.

Major lifestyle changes due to the pandemic affect the mental health of the population, with the manifestation of depressive episodes and anxiety, antisocial behaviors, aggres-

sive attitudes and violence, poor social integration and alcohol addiction, with a higher incidence of cases for men [38]. For students, [39] identifies the psychological pressure, noting cases of severe, moderate and mild anxiety, which are often correlated with the existence of infected relatives or friends, negative economic effects and delays in learning activities. Adequate social support, stable income and the security provided by living with parents are the factors that indicated decreases in anxiety values. The idea is reinforced by [40], who identifies the fear of losing the academic year as the main factor for increased stress and exacerbating psychological suffering, a direct effect of poor e-learning. The psychological pressure of the pandemic on students in Turkey is highlighted by [41] in an investigation of 1704 cases from different universities. It was found that high anxiety is significantly influenced by the variables age, sex, personal income, the existence of cases of COVID-19 in the family or entourage, daily routines, educational environment and problems in social life. Coronavirus syndrome in the Middle East has effects in terms of increasing stress levels for medical students (174 cases), and it was noted that girls have significantly higher values than boys, requiring intervention through psychological support programs during the pandemic [42]. The study presented in [43], conducted in the United Kingdom on a group of 214 students, identified decreases in physical activity and mental wellbeing caused by the pandemic and increases in stress levels and sedentary behavior, with positive and significant associations between the perceived stress level and the sedentary behavior, and suggested that universities should take steps to support students in these difficult times.

Even though online teaching is seen as a success, students often complain about fatigue, difficult work tasks and limited motivation. For teachers, the lack of spontaneity is signaled, as well as the lack of physical interaction that generates an artificial/inauthentic character of communication. Although teachers have quickly assimilated the skills of using work platforms, they perceive a poor quality of interaction through these variants, often being concerned about limiting students' progress. Preparing teachers and students for distance e-learning activities must be a basic element of the strategy of educational institutions, as should rethinking teaching–assessment methods, motivating students, optimizing distance social relationships, reducing inequalities of opportunity and providing high-performance digital services for an authentic communication process [44].

Online activities are a way to combine the act of teaching with the act of learning, where the teacher's actions are focused on the student, who learns autonomously but is guided by the teacher, who answers questions directly and tries to solve the problems reported by students. Freeing students from traditional problem-solving options and providing personalized practical materials helps to streamline teaching, often characterized by overloading teachers' teaching activities [45]. The importance of e-learning in exploratory education for students was studied by [46] through a study of 214 students in China (Fujian) for 3 h/week for 15 weeks. Communication, solving common problems, setting goals, showing confidence in interaction and learning, developing innovative thinking and self-awareness, self-efficacy, diversity of opinion and interpersonal relationships were found to be favorably influenced.

Among the many applications/work platforms used in e-learning during the pandemic, the efficiency and usefulness of the WhatsApp application are highlighted, as it is characterized by simplicity; easy communication; and the ease of sharing Word and PowerPoint files, JPGs, videos and links for learning, and the only major problem is the difficult access to the internet in certain areas and the limited financial resources of students [47]. The learning activity is made more efficient by the use of video materials; the students are more receptive and show a positive attitude towards this type of learning at home, as it is perceived as being more interesting and efficient, facilitating the understanding of the transmitted information [48]. The adaptation of university activities to the realities of the pandemic required video conferencing teaching (WVC); the quality of teaching was appreciated as good by 82% of students surveyed (162 subjects), video sessions were

attractive and challenging in terms of intellectual demands and the attitude of teachers encouraged student participation [49].

The opinions of 40 students from Megarezky University (Indonesia) during the academic year 2019/2020 are analyzed by [50], indicating that prior to the pandemic, online teaching activities were not used. The best application for e-learning is considered to be WhatsApp, which offers a positive perspective on online education, the platform being cheap, accessible and efficient, with multiple learning facilities (chat, video call, voice notes). The government, the universities and the professors should take steps to facilitate internet access and provide financial support to students. Socializing online on Facebook is a favorite option for young people, as they are familiar with this type of communication, but the elderly are skeptical about this option, and their familiarization with digital technologies is part of public policy [51]. Even if students use the Facebook social network frequently or even a few hours a day, it is not seen as a learning or research option, but only as a way to communicate, exchange information with friends and spend free time [52].

The mere provision of teaching materials is not enough to obtain favorable results from students. An immediate/instantaneous feedback from the teacher has an important role, but attention must also be paid to the students' feedback in order to facilitate the communication process. Live broadcasting is more effective than recorded video because it increases students' academic performance [53]. The need for effective and timely feedback for Korean physical education lessons is supported by [54]. The teachers need to be well trained and familiar with the online environment, the universities need to provide technical support and students need to receive encouragement and objective assessments.

A study involving 476 students in Bangladesh identified moderate and severe depression in 15% of students and anxiety in 18.1% of cases, with older students suffering from more severe depression. The causes are related to financial problems, the state of academic uncertainty and the problematic internet connection for those in remote areas. Needs related to scholarships for internet access and ensuring a family-friendly and pressure-free climate are reported by [55]. The technological gap and differences in internet access between students in urban vs. rural areas in Bangladesh are highlighted by [56]; as these aspects influence their behavior and accessibility to information, reducing the gaps in digital technology in developing countries is a priority. Limited performance and low student motivation (282 cases) for online activities are noted by [57], the reasons being related to the deficient infrastructure that cannot optimize the teaching/learning process, i.e., the problems with the internet connection and the access to the e-learning platforms.

## 5. Conclusions

Applying this questionnaire to students identifies a multitude of favorable features that support the promotion of online teaching activities in the future but allows us to identify the many shortcomings in the teaching–learning–assessment act, which make us regard more reluctantly this teaching type imposed by the pandemic context for the university education. The fact that there are differences in the average scores for opinions between the sexes and between the students' living areas that are not statistically significant except in some cases facilitates the identification of common opinions and nuanced differences between the groups investigated, which allows adapting the teaching technology and eliminating the weak spots signaled by the students at the moment, allowing the efficiency of e-learning in the future. The main aspects noted can be summarized as follows: the attractiveness and superior socialization identified for those in urban areas; higher participation in activities and a higher concern for teaching activities identified for girls; technical problems and difficulties in accessing the platform, exacerbated by a lower level of computer skills, for those in rural areas; and the significant increase in free time for girls and students in urban areas. Even in these cases, however, the effect size ($\eta^2_p$) still indicates low influences of independent variables on dependent ones, the values <0.05 indicating a low effect and the values < 0.01 indicating a zero effect, so the few cases of significant differences are not confirmed by strong practical effects.

*Limits of the Study*

The data collected and interpreted cannot be generalized, firstly because they are the result of the opinions of a relatively small group and secondly because the investigated specializations (physical education and sports and physiotherapy) have their own defining characteristics in the academic world, representing a niche segment, where the theoretical activities (possible to be easily transferred online) must be complemented with practical activities (where the biggest problems in conducting lessons and achieving the competencies in the subject sheets appear). The application of the questionnaire for other faculties and stages of study (master's and doctorate) in different university centers would allow a thorough investigation of the viability of online teaching technologies and the identification of common features and inherent differences, conditioned by the students' specializations, between the specific areas and socioeconomic development levels of the respective university centers. Last but not least, starting a study not only to investigate and present the opinions of teachers related to the methodology and efficiency of the online teaching process but also to identify the problems in their adaptation would be beneficial, thus enabling the development and selection of viable solutions for the reform of trainers as a premise for improved performance in higher education in the context of COVID-19.

**Author Contributions:** Conceptualization, G.D.M. and G.M.; methodology, G.D.M. and L.G.; software, G.D.M., I.S. and G.M.; formal analysis, L.G.; investigation, G.D.M. and G.M.; resources, G.D.M. and L.G.; data curation, G.D.M. and I.S.; writing—original draft preparation, G.D.M.; writing—review and editing, I.S. and G.M.; visualization, G.D.M. and G.M.; supervision, I.S. and L.G.; project administration, G.D.M. and G.M.; funding acquisition, L.G. All authors have read and agreed to the published version of the manuscript.

**Funding:** This research received no external funding.

**Conflicts of Interest:** The authors declare no conflict of interest.

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
