# Peer review of "Investigating the Attitudes of First-Year Students of the Faculty of Physical Education and Sports of Galati towards Online Teaching Activities during the COVID-19 Pandemic"

_applsci, doi:10.3390/app11146328_

Round 1
Reviewer 1 Report
I find the subject of the paper very interesting and necessary; it would be great to expand the sample. Here are some comments that I think may help:
- The study design should be specified in the methodology and sampling.
- Combine line 166 with what is included in the following paragraph (line 175-179), as putting that in there without giving more information seems confusing to me.
- Results:
- You could start by describing the characteristics of the sample (they have been included in methods).
- Tables 4,6,8, and 10 seem confusing to me, please modify them. And I think figures 1-4 do not add anything so I would remove them (that information is already in the tables). If you leave them, it would be good to mark the significant differences.
- As for the discussion I find it very scattered, I think it should focus more on what is related to the findings of the study. It would be good to start with a paragraph highlighting the results obtained before stating consistency with other papers.
- The final "Recommendations" paragraph could be included in the discussion in the form of limitations.
- On the other hand, I believe you have already published the internal consistency data, in which case I do not see the need to include them in the results. It would be enough to put in methodology that the questionnaire has xx.
Author Response
I find the subject of the paper very interesting and necessary; it would be great to expand the sample. Here are some comments that I think may help:
- The study design should be specified in the methodology and sampling.
- Answer: Thanks you for the comment. The study was designed for cross-sectional research, including first-year students. The methodology used was the standard one commonly used for attitudinal investigations
- Combine line 166 with what is included in the following paragraph (line 175-179), as putting that in there without giving more information seems confusing to me.
- Answer: Thanks for the comment. We have introduced the bibliographic source that refers to the previously published article - source no. 37.
- Results:
- You could start by describing the characteristics of the sample (they have been included in methods).
- Answer: Thanks for the comment. We have included the description of the sample in the methodology in the previous article - see reference no. 37.
- Tables 4,6,8, and 10 seem confusing to me, please modify them. And I think figures 1-4 do not add anything so I would remove them (that information is already in the tables). If you leave them, it would be good to mark the significant differences.
- Answer: Thanks for the comment. The mentioned tables condense 3 types of information: Likert scale and percentage values on scores, data of variable 1 (gender) variable data 2 (average of origin). We have thus reduced the volume of information that would have required 8 more tables with repetitive information on items. The graphs help to make it easier to present the results, considering that reading the tables is more difficult. Significant differences are listed in red in the tables.
- As for the discussion I find it very scattered, I think it should focus more on what is related to the findings of the study. It would be good to start with a paragraph highlighting the results obtained before stating consistency with other papers.
- Answer: Thanks for the comment. In the chapter on discussions, we have completed a paragraph that contains information strictly from the study.
- The final "Recommendations" paragraph could be included in the discussion in the form of limitations.
- Answer: Thanks for the comment. We have modified the chapter Recommendations in the Limits of the study by adding other additional information.
- On the other hand, I believe you have already published the internal consistency data, in which case I do not see the need to include them in the results. It would be enough to put in methodology that the questionnaire has xx.
- Answer: Thanks for the comment. For the validation of the first two working hypotheses in the study, table no. 1 was attached, which includes the values related to the internal consistency and the variability of the answers. We have included this table to facilitate the reading of this study in the situation where the first article - reference no. 37 will not be consulted.

Reviewer 2 Report
As you say in the 'Recommendations' section, it is not possible to conclude on a general hypothesis with a finite population, it is important to delimit the study to the studied area. Therefore, it is important that you delimit the study to the faculty in which you focus the study, so you should add to the title that the faculty studied is the Faculty of Physical Education and Sports of Galati.
Once you make this change you should go to section '2.1. Participants' and justify why the sample size is significant. For example, with a sample size formula for finite population.
And, finally, delete the 'Recommendations' section, replace it with a new paragraph at the end of the 'Conclusions' section that talks about the new limitations and includes deductions for future research.
As for the rest of the content, I do not have any changes to recommend, it makes a good literary review to support the relevance of the problem to be studied and a good structuring of the content, it uses the correct methodology for this type of study and it is a consistent and well-detailed methodology to give significance to the results they show, makes a good discussion of the results with respect to the studies carried out previously, and marks the conclusion obtained well.
Once the authors have made these changes, I think the article is publishable.
Author Response
As you say in the 'Recommendations' section, it is not possible to conclude on a general hypothesis with a finite population, it is important to delimit the study to the studied area. Therefore, it is important that you delimit the study to the faculty in which you focus the study, so you should add to the title that the faculty studied is the Faculty of Physical Education and Sports of Galati.
Answer: Thanks for the comment. The title has been modified according to your recommendations.
Once you make this change you should go to section '2.1. Participants' and justify why the sample size is significant. For example, with a sample size formula for finite population.
Answer: Thanks for the comment. The study was designed for cross-sectional research, including only first-year students. The questionnaire was applied to a percentage of 83.05% of the total undergraduate students in the first year, respecting the proportion between genders.
And, finally, delete the 'Recommendations' section, replace it with a new paragraph at the end of the 'Conclusions' section that talks about the new limitations and includes deductions for future research.
Answer: Thanks for the comment. We have modified the chapter Recommendations in the Limits of the study by adding other additional information.

Round 2
Reviewer 1 Report
Thanks for the responses and changes, I think the manuscript has improved.
Still I think the methodology and sample characteristics could be better described in the text, so that I don't have to refer to the previous article.